# A Graph Neural Network with Spatio-Temporal Attention for Multi-Sources Time Series Data: An Application to Frost Forecast [note 1]

**DOI:** 10.3390/s22041486

**Published:** 2022-02-15

**Authors:** Hernan Lira, Luis Martí, Nayat Sanchez-Pi

**Affiliations:** Inria Chile Research Center, Las Condes 7550268, Chile; lmarti@inria.cl (L.M.); nayat.sanchez-pi@inria.cl (N.S.-P.)

**Keywords:** frost forecasting, graph neural networks, spatio-temporal attention

## Abstract

Frost forecast is an important issue in climate research because of its economic impact on several industries. In this study, we propose GRAST-Frost, a graph neural network (GNN) with spatio-temporal architecture, which is used to predict minimum temperatures and the incidence of frost. We developed an IoT platform capable of acquiring weather data from an experimental site, and in addition, data were collected from 10 weather stations in close proximity to the aforementioned site. The model considers spatial and temporal relations while processing multiple time series simultaneously. Performing predictions of 6, 12, 24, and 48 h in advance, this model outperforms classical time series forecasting methods, including linear and nonlinear machine learning methods, simple deep learning architectures, and nongraph deep learning models. In addition, we show that our model significantly improves on the current state of the art of frost forecasting methods.

## 1. Introduction

Generating accurate weather forecasts from reliable localized data is a key feature of precision agriculture that enables farmers to improve their resources in terms of efficiency, productivity, sustainability, etc. Moreover, it is a tool for countering weather uncertainty by reducing the risks posed by extreme weather that can impact the overall quality of the production. Frost is one such threat that kills plant tissue, causing low production and economic losses since it prevents the normal development of crops. During some periods of the year, temperatures can drop considerably between day and night, with temperatures reaching below freezing. A low temperature can cause the crop to flower early or if there is frost it can cause a considerable reduction in production [1]. These negative consequences could be prevented or mitigated with a frost forecast model that provides information to the farmer regarding the probability of a frost event several hours in advance, so the farmer can take action to protect the crops.

From a forecasting perspective, the prediction of frost events presents some challenges. For instance, they are a complex meteorological phenomenon influenced by a combination of environmental factors, including air temperature, humidity, radiation, and wind, and local factors, including topography and field orientation. Since there are local factors involved, frost events can occur in small areas even within the same crop field [2]. Therefore, a method to collect high-resolution weather data is required.

Typical weather data sources such as open weather, used for global weather forecasts, or national weather stations are useful to understand the weather dynamics across big areas. Previous research used frost forecast models from these sources, but they are limited to forecasting in the vicinity of the weather station [3]. Therefore, these methods do not provide the resolution required when seeking to forecast the weather in a specific location/field. As a consequence, farmers use these forecasts as a reference, but most of their decisions to prevent frost events are based on experience and intuition.

To collect data specifically from a field, we developed a low-cost and easy-to-install IoT platform, with environmental sensors and cloud technologies for collecting and storing data. Specifically, we measure air temperature, air humidity, and relevant metadata. In addition, we use meteorological data collected from weather stations located near the field [4]. The purpose is to provide the farmer with relevant information so that they can make informed decisions. The frost forecasting model requires an intelligent component that uses field data to capture specific conditions about the field combined with weather stations data to capture weather dynamics and possible future scenarios.

The majority of research relating to frost forecasting is based on simulating partial differential equations or traditional statistical models to predict future weather conditions [5]. This approach is computationally expensive as it requires a recurrent theoretical upgrade to incorporate weather and atmospheric assumptions. On the contrary, machine learning algorithms do not make any assumptions about weather behavior. Instead, they use historical weather data as an input and train a model to predict future weather values [3].

There are some challenges related to machine learning models. Since frost can be highly variable across a small area, the collection of temperature data usually from weather stations is not available with sufficient frequency. In addition, the number of frost events during the year is relatively small, making it difficult to build an accurate prediction model due to limited available data [2]. Finally, by viewing the model as a binary classification problem, i.e., Frost/No Frost, we need to consider that both errors are unwanted. If no frost is predicted and the frost occurs, it may impact on the partial or total loss of production. On the contrary, if frost is predicted and the frost does not occur, unnecessary resources such as fuel and electricity used to mitigate the frost will be wasted.

In light of the aforementioned constraints, especially the scarcity of data and small datasets, a range of machine learning models were evaluated, including models to learn time series data as well as advanced deep learning architectures. In particular, graph neural networks (GNN) and attention mechanisms were considered suitable for this problem, since they incorporate spatial knowledge that can model field and environmental interactions [6,7]. In addition, since the occurrence of frost is caused by a prior movement of environmental factors, GNNs can be naturally extended to model this type of temporal interaction. Therefore, in this paper we discuss a time series forecasting problem using GNNs and attention.

In this study, we collected air temperature and humidity data from an experimental site and from 10 weather stations. In particular, we propose GRAST-Frost, a GNN with spatio-temporal attention architecture for frost forecast. We map weather stations’ locations to nodes on a graph and construct the edges based on geographical proximity. Furthermore, the adjacency matrix is optimized during the training phase, therefore other interactions can be learned. We utilize spatio-temporal attention to incorporate similar locations and time.

To the best of our knowledge, deep learning and graph neural networks were so far not applied to the frost forecasting problem. Although considerable research was devoted to this area, most of the research focused on developing an IoT platform for collecting in-place weather data. Consequently, little attention was paid to developing a model that can take advantage of multiple data sources and spatio-temporal dynamics. However, GNN models were recently adopted for traffic forecasting, epidemiology, and fraud detection. The implementation of GNN models in these scenarios showed the potential for predicting multivariate time series [8,9] and graph attentional networks for capturing spatio-temporal dynamics [7,10,11].

Our contributions to this field are two-fold. First, we propose a full pipeline for the frost problem, including the development of an IoT platform, data collection, and forecasting model using two data sources. Second, we approach the frost problem by proposing a multivariate time series forecasting method, GRAST-Frost, that computes each time series at the same time. This method is based on a graph data structure, and it uses a spatio-temporal attention mechanism for weighted relevance according to time and/or space.

Moreover, this paper seeks to answer the following research questions:Is a GNN capable of improving the time series forecasting of data sources from different locations in comparison to state-of-the-art frost forecasting methods?Does the spatio-temporal attention mechanism improve forecasts?Does the combined use of different data sources improve forecasts?

To answer the first question, we compare our approach with classical time series, machine learning, and deep learning methods by classification and regression metrics, which are currently the state of the art for frost forecasting. With regard to the second question, we compare our approach with GNN models proposed in previous studies that do not use spatio-temporal attention mechanisms. For the final question, we compare the forecasts using separate data sources.

The proposed model outperforms current state-of-the-art methods for predicting temperature and classifying frost from 6 h, 12 h, 24 h and 48 h in advance. Furthermore, graph-based modeling and spatio-temporal attention mechanisms are key factors to perform a more accurate prediction and minimize classification errors.

The rest of the paper is organized into the following sections: Section 2 examines previous studies regarding machine learning for time series, weather, and frost forecasting. Section 3 describes the platform and our proposed method. After that, Section 4 presents the experiments and results, and then we offer a discussion of the results in Section 5. Finally, Section 6 concludes the paper and presents possible directions for future work.

## 2. Related Work

Mort et al. [12] and Verdes et al. [13] are two research studies that address the frost phenomenon using machine learning. In these studies, the authors use artificial neural networks to create temperature prediction models based on weather time series data and apply them to agricultural applications. Their principal objectives are to predict the next day’s minimum temperature using historical data. In recent years, there were great advances in the field of internet of things (IoT) systems and deep learning. Thanks to these advancements, it was possible to install a wide variety of sensors and collect data from almost any place of interest with the purpose of building accurate prediction models. Regarding IoT systems designed for weather forecasting, Muck et al. [14] designed an IoT based weather station using a Raspberry Pi, which provides short-term weather forecasting. Similarly, Levin et al. [15] presents a weather forecasting system based on a Raspberry Pi 3 Model B+ with environmental sensors and a weather forecasting algorithm. Their systems monitor air temperature, humidity, pressure, and altitude at experimental locations. Their weather forecast algorithms are based on a linear regression model. Other studies such as Diedrichs et al. [3] and Castaneda–Miranda et al. [16] used IoT devices to extract weather data from selected locations as well, but instead of focusing on weather, they used classic applied machine learning techniques to predict frost events. Likewise, the research group of Guillén–Navarro et al. [1,17] developed over the years an IoT platform to predict frost events. This platform appears to be more robust than the previous studies in terms of engineering and technological components. Although these studies made interesting progress in terms of IoT and sensor data collection systems, the development of their machine learning models was limited. Therefore, the resulting prediction results are unsatisfactory in terms of error rates and/or classification metrics. In addition, the data sources were constrained to the experimental field where the system is located.

Few studies attempted to focus on the development of the frost forecast model itself. For example, Ding et al. [2] concentrated their efforts on the development of a causal-effect machine learning model that uses locally collected temperature, humidity, and radiation data to create frost prediction. They were able to describe causal relationships between variables and outputs; however, their model requires improvements to minimize the false-positive predictions. Another example is the study by Cadenas et al. [18], which was based on a soft computing framework that collects and stores weather data. They propose a data preprocessing technique to build fuzzy time series from raw data and serve it as an input to classification and regression problems. In addition, Guillén–Navarro et al. [19] used a simple long short-term memory (LSTM) architecture to produce frost forecasts from data collected using their IoT system. Although these studies provide interesting methods to address the frost forecasting problem, there is still a wide range of solutions to investigate. For example, exploring current developments within deep learning models that could improve forecasts and include different data sources.

In contrast to frost forecasting, weather-related forecasting has plenty of studies that use advanced deep learning techniques. For instance Shi et al. [20] proposed a fully connected convolutional LSTM network to predict short-term future rainfall intensity in a local area and extract spatio-temporal dynamics of the data. Likewise, Mehrkanoon et al. [21] proposed a model for predicting temperature and wind speed 1 to 10 days in advance using a convolutional neural network. They introduced an architecture based on 1D-CNNs to process tensor 3D data and to extract spatio-temporal relations. In addition, Hewage et al. [5] presented a weather forecasting model that uses an LSTM and a temporal convolution network. The results obtained are better than classical time series forecasting and classical machine learning. However, these models do not capture the complexities of our specific problem. For instance, we need to deal with multivariate time series forecasting of several time series at different locations. In the aforementioned approaches, there are no spatial relations between the entities (e.g., different cities), the interaction is determined by the entity order in the tensor. To capture the spatio-temporal dynamics of different entities, a promising approach is to use GNNs, which have the capacity to model temporal dynamics of nodes and spatial dynamics between them at the same time.

GNNs showed recent progress in the area of time series forecasting and spatio-temporal relations. Moreover, a GNN can extract greater insights compared to that of networks that can only analyze data in isolation. This is achieved by obtaining structural relationships between the data [22]. There are a number of domains in which GNNs were successfully applied in recent years, such as traffic flow forecasting, fraud detection, epidemiology, and forecasting weather-related events. In regards to the latter, Wilson et al. [23] addressed the spatio-temporal correlation in the data by proposing a deep learning model based on a weighted graph convolutional LSTM. The general goal was to capture temporal autocorrelation with the LSTM and the spatial relationships with the graph convolution. Similarly, Khodayar et al. [24] presented a spatio-temporal Graph convolutional network (GCN) for short-term wind speed forecasting. Another example is the study by Wang et al. [25], which proposed a graph-based model to predict PM2.5 particle concentration and capture the spatio-temporal dependencies.

The most recent advances in GNN were applied to other domains. For instance, a study by Cheng et al. [7] presented a model using GNN for fraud detection in credit card transactions. They implemented a spatio-temporal attention mechanism which produces the input for a 3D convolution network. As per the previous study, Gao et al. [26] proposed a GNN with spatio-temporal attention mechanism and a GRU architecture to forecast the number of infected cases in a pandemic by considering local disease status and demographic and transmission dynamics. There were several GNN models applied to traffic flow forecasting and urban planning. In particular, Song et al. [27] developed a spatio-temporal GCN with a synchronous temporal mechanism to predict the flow of a network. Likewise, the studies by Lu et al. [28], Kong et al. [10] and Li et al. [11] proposed different model versions of a spatio-temporal GNN with attention mechanisms for urban sensor value forecasting, traffic flow forecasting, and segment-level traffic prediction, respectively.

The goal of this study is to predict frost events by using air temperature and humidity data obtained from an IoT system installed on an experimental field and weather stations located around the field. We have two main sources of inspiration. First, we are inspired by recent advancements in GNN models as detailed previously, especially the spatio-temporal attention mechanism. Second, we are inspired by the study of Wu et al. [9], which is a model for multivariate time series forecasting using a GNN, and by the recent study of Shang et al. [8], who proposed a model for multivariate time series forecasting using a GNN in which they consider pairwise interactions between features in a node representation. Therefore, the contribution of this paper is a multivariate frost forecasting model based on a GNN with a spatio-temporal attention mechanism.

## 3. Proposal

In this paper, we propose a bivariate time series forecasting model to predict the occurrence of frost. The benefit of such a model is that it can be trained with historical time series data $X=\{x_{t_{1}},x_{t_{2}},\ldots ,x_{t_{m}}\}$ with $x_{t}\in R^{2}$ the value of the bivariate variable at time *t* is used to forecast future values of the variable in a certain time-window *r*, $Y=\{x_{t_{m+1}},x_{t_{m+2}},\ldots ,x_{t_{m+r}}\}$. Then, the goal is to create a mapping function from *X* to *Y* and minimize the loss, typically using a l2 regularization [9].

In particular, given a set of input data P={n,temp,hum} and a derived label F={1,0} which indicates the presence (1) or absence (0) of frost for each one of the records. We aim to forecast the minimum temperature and the frost class for future time windows $\left\{t_{m},\ldots ,t_{m+r}\right\}$ based on the historical time window $\left\{t_{1},\ldots ,t_{m-1}\right\}$.

In addition to bivariate time series forecasting, we model the spatial and temporal relationships of weather data from multiple locations. For that purpose we utilize GNNs to describe and formalize those relationships. The following are important definitions for graph modeling [9].

*Graph:* a graph is represented as G=(V,E) where *V* represents the set of nodes and *E* represents the set of edges. There are *n* number of nodes in a graph.*Node neighborhood:* describes a set of nodes connected by an edge. A singular node v∈V and an edge e=(v,u)∈E maps from *v* to *u* describes the connection between nodes. The neighborhood of *v* is defined as N(v)=u∈V∣(v,u)∈E.*Adjacency matrix:* states the connections between nodes in a graph. It is denoted by a matrix $A\in R^{n\times n}$ with $A_{ij}=q>0$ if $(v_{i},v_{j})\in E$ and $A_{i,j}=0$ if $(v_{i},v_{j})\notin E$.

Then, the graph network is formally defined as G=(V,E,A), which represents the relationships between nodes in the spatial dimension.

### 3.1. Data Sources

The model was trained and tested using data collected from our IoT platform and 10 meteorological stations, which are all located in the central region of Chile. Appendix A lists the stations and their geographical location.

The 10 meteorological stations, which are within close proximity of the orchard, provide structured temperature and humidity data in the form of a graph. Here, the data are collected every hour. The 10 meteorological stations considered for this study and their representation as a graph are shown in Figure 1.

The IoT platform was developed with both air temperature and air humidity sensors and consists of 12 low-power wireless sensor nodes (motes). The latter are divided into eight sensor data nodes and four repeaters, which are connected to a gateway through a SmartMesh IP manager. The wireless sensor network is exposed directly to environmental conditions (sun, dust, rain, and snow), and therefore all sensors are protected by an Internal Protection 65 (IP65) rating enclosure. The IoT platform was installed in a local orchard where data were collected every 10 s at four different heights above ground (one, two, three, and four meters). An image of the IoT platform in situ in the orchard is shown in Figure 2.

Data were collected between 4 September 2020 until 5 April 2021. Training data were taken from the months of September to February and the last two months of data were used for training. It is assumed that environmental factors can have an accumulated impact on frost; therefore, the prediction was performed using time series data. For this reason, we need the model to learn time-related patterns, which is why we do not randomly split training and testing data.

All the data are preprocessed for missing values and outliers. Data from the IoT platform is downsampled using 1-min time-windows. In addition, to map the frequency of both data sources, data from weather stations are linearly interpolated. Finally, in case of the classification of frost, a label is created in the training data with two classes (Frost/No Frost) and given the imbalance between them, the synthetic minority oversampling technique (SMOTE) method was applied.

### 3.2. Forecasting Model

In this study, we develop a GRAST-Frost model, the architecture of which is shown in Figure 3. First, we convert our input *P* into a 3D (spatial, temporal and features), high-order tensor representation X. The tensor is then fed into the graph neural network, which aims to correlate the IoT platform data with that of the meteorological stations. For each *t*, we create a graph of nodes that relate to each of the stations (meteorological and the IoT platform) and their corresponding edges. Then, we apply a spatio-temporal attention mechanism and a 3D convolution layer to obtain a feature-learned tensor $X^{C}$. With spatio-temporal attention and 3D convolution, the idea is to weigh the importance of different dimensions and find hidden patterns from the input data. Finally, $X^{C}$ is fed through a recurrent neural network to produce a forecast with two different loss functions depending on the type of task. We present details of each part of this architecture in the following subsections.

To forecast local weather conditions at the orchard using the aforementioned sensory data, we predict at differing time intervals into the future (6, 12, 24, and 48 h). At a single time step, the geographical locations of the nodes are graphed as can be seen in Figure 1, where the blue dots represent the meteorological sites and the green dot depicts the experimental site. For multiple time steps, the graph is expanded into a spatial-temporal graph where feature values for a given node are related to its previous and future values and its spatial neighbors. A schematic view of this idea is shown in Figure 4.

#### 3.2.1. Feature Engineering

To represent the input data as a tensor $X\in R^{T\times S\times F}$ where *T*, *S*, and *F* denote the temporal, spatial, and feature dimensions. In particular, for each spatio-temporal pair composed of a certain time horizon and location, a feature vector is built ($f\in R^{F}$) based on the measurements collected in that pair. There are a total of T×S spatio-temporal pairs, each one of them are features of *F* dimensions. Based on [7], the feature vector is composed of two parts: feature measurements and graph features. For measurements, we include raw temperature, humidity data, mean, standard deviation, median, maximum, and minimum values for a specific time horizon. For the graph related features, we include several metrics obtained from the graph neural network processing described in the following section.

#### 3.2.2. Graph Construction at a Single Time Step

Given the tensor *X*, the geographical location (latitude, longitude) of the weather stations and experimental field is obtained. This data are used to calculate the geographical proximity, which in turn contributes to the development of the adjacency matrix. Nodes are simply a subset of the tensor in time *t*. We utilize an aggregation function to reduce the edge updates to a single element. Therefore, for a single node, we summarize the interactions with other nodes.

The nodes in the graph G(V,E) are constructed to represent input data, by associating features such as (temp,hum) and location (lat,lon). In total we have 11 nodes. Generally, we have two types of nodes: *v* is the node corresponding to the experimental field, and *u* is the node corresponding to the weather stations. Thus, $f_{v}$ and $f_{u}$ are the corresponding feature vectors for each one of them.

The edges are constructed based on geographical distance between nodes. We construct a weighted graph where the weight is inversely proportional to distance. Formally, in a weighted graph G=(V,E,w) [6], that do not contain any cycle of negative weight, the distance between node *u* and *v* is defined as d:V×V→R as
(1)$$\begin{array}{cc} d(u,v)=0 & \mathrm{if}\;u=v,\;\mathrm{and}\\ d\left(u,v\right)=\|\left(location_{u},location_{v}\right)\| & \mathrm{otherwise}, \end{array}$$
that corresponds to the Euclidean distance between the location of nodes *u* and *v*.

Therefore, the weight is,
(2)$$w_{uv}=\frac{1}{d(u,v)}$$

For this study, all nodes are integrated, thus, there are edges between all the nodes. Therefore, the feature vector of the edge is $f_{e}=w_{uv}*f_{gnn}$, which is the weighted measurement of $f_{gnn}$, features to be obtained by the graph neural network (GNN).

#### 3.2.3. Spatial-Temporal Graph Construction

To represent the relationship between each node and its neighbors across time, inspired by [27], we connect all nodes with themselves sequentially for each time step. This allows us to create a spatial-temporal graph by sequentially connecting nodes from previous to current and future time steps as shown in Figure 5. Therefore, it is possible to understand the relationships between nodes through time. In practice, we create a new adjacency matrix $A^{T}\in R^{n\times T}$ for the spatial-temporal graph. The new adjacency matrix can be formulated simply as
(3)$$A_{i,j}^{T}=\left\{\begin{array}{cc} 1 & \mathrm{if}\;v_{i}\;\mathrm{is}\;\mathrm{related}\;\mathrm{with}\;v_{j},\\ 0 & \mathrm{otherwise}. \end{array}\right.$$

As illustrated in Figure 5 the new adjacency matrix has dimension N×T, where its diagonal represents the adjacency matrix for a single time step and a relation of the nodes through time.

For each time step a GNN consumes the graph described on the previous section. The information of interactions between nodes is described in terms of nodes involved u,v, time *t*, location *l* and measurement *m*. Based on it, the edges are constructed. Features for nodes $f_{v},f_{u}$ and edges $f_{e}$ are initialized accordingly. Then, we iteratively include graph constructed in subsequent time steps by updating the representation of nodes and edges based on the previous time step and weighted by the adjacency matrix $A^{T}$. Edge features are updated following
(4)$$f_{euv}^{t}=FC_{e}\left(f_{euv}^{t-1},id_{u},id_{v},A_{t,t-1}^{T}\right)\;,$$
where $f_{euv}^{t}$ is the edge feature between nodes *u* and *v* in time *t*, $FC_{e}$ is a fully connected neural network with a ReLU activation function which computes all the edges updates. In addition, $id_{u},id_{v}$ are the location on graph of node *u* and *v*. Then, for each time step we proceed to update the node features $f_{u}$ and $f_{v}$. For each node, we sum all the edges that are connected to that node. Regarding $f_{v}$ the equation is
(5)$$f_{v}^{t}=FC_{v}\left(\sum_{i}^{N}f_{e_{i}}^{t},f_{v}^{t-1},A_{t,t-1}^{T}\right)\;.$$

Here $f_{v}^{t}$ is the node feature in time *t*, *N* is the total number of nodes and $FC_{v}$ is a fully connected neural network.

Finally, we calculate the feature vector of the graph $f_{g}$ which is the average of all the updates of $f_{e}$ and $f_{v}$ for every time step.

The output of the GNN are the updated values of $f_{v}$, $f_{u}$, $f_{e}$ and $f_{g}$. We use these values to complete the feature vector *F* on the tensor X regarding the graph part, as described in the previous section.

#### 3.2.4. Spatio-Temporal Attention Mechanism

The idea of the attention network is to weigh the importance of spatial and temporal values from the current measurement for a specific node and time. We use the GAT model [6] to extract Spatio-temporal similarity features. The idea is to update the embedding information of each node using the aggregate data from its neighbors. Therefore, weather stations and the IoT platform receive prior temperature and humidity data from nearby areas. This allows us to hypothesize that a specific prediction is more likely due to a higher importance being given to meteorological sites in close proximity to the orchard, rather than those located further away, and recent measurements being given a higher priority compared to that of older measurements. Given the feature tensor X, we can query the temporal values $\forall t\;X_{tsf}$ (t∈{1,2,...,T}) to extract the time horizon, and the spatial values $\forall s\;X_{tsf}$ (s∈{1,2,...,S}) to extract the location coordinates.

##### Temporal Attention

In the temporal dimension, there exist correlations between temperature and humidity values in different time steps. Since weather is highly dynamic, temporal correlations are variable under different conditions. Therefore, to have a mechanism for capturing those correlations, we use attention to adaptively obtain the importance of previous data in relation to current data.

Based on [7], considering a tensor X, the temporal attention layer is the temperature and humidity values for a specific time multiplied by a weighted sum of the matrix representation of all temporal values. Formally, it is described as
(6)$$X^{t}=\sum_{t=1}^{T}a_{t}X_{tsf}\;\mathrm{with}$$
(7)$$a_{t}=\frac{\exp(\left(1-\lambda_{t}\right)\cdot FC_{t}\left(W_{t},X_{tsf}\right))}{\sum_{t=1}^{T}\exp\left(\left(1-\lambda_{t}\right)\cdot FC_{t}\left(W_{t},X_{tsf}\right)\right)}\;.$$

Here $a_{t}$ is the weight for each time step, $FC_{t}$ is a fully connected layer with ReLU activation and the weight vector $W_{t}$; $\lambda_{t}\in \left[0,1\right]$ is the temporal penalty factor to control the importance of temporal attention; $X^{t}$, the output of the temporal attention layer is a tensor with $X\in R^{T\times S\times F}$.

##### Spatial Attention

The spatial dimension, temperature, and humidity values from different nodes have varying influences, and due to the weather, the behavior of these influences are highly dynamic. In particular, we are interested in capturing correlations between the nodes on the weather values in the spatial dimension. Therefore, we want to explicitly capture the relationships between close and distant nodes.

Given the output from the temporal attention layer $X^{t}$, the spatial attention mechanism is applied. Formally, it is described as
(8)$$X^{st}=\sum_{s=1}^{S}a_{s}X_{tsf}^{t}\;\mathrm{with}$$
(9)$$a_{s}=\frac{\exp(\left(1-\lambda_{s}\right)\cdot FC_{s}\left(W_{s},X_{tsf}^{t}\right))}{\sum_{s=1}^{S}\exp\left(\left(1-\lambda_{s}\right)\cdot FC_{s}\left(W_{s},X_{tsf}^{t}\right)\right)}\;,$$
where $W_{s}$ is the weight of the fully connected spatial network $FC_{s}$, and $X^{st}$ is the output of both self-attention layers. The output is reshaped into a tensor format with the same order as X; $a_{s}$ is the weight for each spatial step; and $\lambda_{s}\in \left[0,1\right]$ is the spatial penalty factor to control the importance of spatial attention.

#### 3.2.5. Convolution Process

Several studies stated the benefits of applying convolutional neural networks to feature tensors based on GNN modeling [29,30,31]. In our case, we used a 3D convolutional network with the $X^{st}$ tensor as an input. The idea is to extract hidden patterns from spatio-temporal features by stacking multiple layers in the architecture.

The 3D convolution is represented as
(10)$$X_{k}^{c}=\sum_{k}X^{c-1}\left(d_{t}-c_{a},d_{s}-c_{b},d_{f}-c_{q}\right)K_{i}^{l}\left(c_{a},c_{b},c_{q}\right)\;.$$

Here, $K_{i}^{l}$ is a 3D kernel in the lth layer and ith kernel in a convolution with feature $X^{c-1}$. In particular, the 3D convolution kernel is ($c_{a}$, $c_{b}$, $c_{q}$). The first layer of $X^{c-1}$ is the output of our attention mechanism $X^{st}$. $d_{t}$, $d_{s}$, and $d_{f}$ are the dimensions of $X^{st}$ considering temporal, spatial, and feature components, which equals to T,S,F of the first convolution layer.

Finally, the output feature $X^{c}$ is
(11)$$X^{c}=\sigma \left(\sum_{k}X_{k}^{c}+b^{c}\right)\;,$$
where $b^{c}$ is the bias parameter, and σ is the sigmoid function.

#### 3.2.6. GNN Forecasting

In the last part of the forecasting model, we apply a recurrent neural network to capture the sequential aspect of the problem and produce forecasts based on historical data. Given the tensor $X^{c}$ we use a sequence-to-sequence model (seq2seq) [32] over each node, i.e., $\forall s\;X_{tsf}^{c}$. Thus, we extract the transformed series from the experimental field and weather stations. The reason for using seq2seq is that in a graph structure, we can perform recurrent graph convolution to handle all series simultaneously [8]. In practice, for each series we used $\left\{t_{1},...,t_{m-1}\right\}$ time values to train the model and $\left\{t_{m},...,t_{r}\right\}$ to forecast the weather using r∈{6,12,24,48} h in the future. Specifically, for each time *t*, the seq2seq model takes $X_{tsf}^{c}\;\forall s$ for all series and updates the internal hidden state from $H_{t-1}$ to $H_{t}$. The encoder recurrently updates the training data to be included, producing $H_{t+r}$ as a summary. The decoder takes that input and continues the recurrence to include all the testing data for the forecasting phase.

Finally, we use two loss functions, one for classification and one for regression. The former loss function is defined as
(12)$$L=\frac{-1}{N}\sum_{i=1}^{N}\left[y_{i}log\left(H_{i}\right)+\beta \left(1-y_{i}\right)log\left(1-H_{i}\right)\right]\;,$$
where *N* is the total number of samples in the series, β is a sample weight regarding the distribution of Frost/No Frost; $y_{i}\in \left\{0,1\right\}$ is the real label and $H_{i}$ is the value score produce by the forecasting.

The regression loss function uses mean squared error
(13)$$L=\frac{1}{N}\sum_{i=1}^{N}y_{i}-H_{i}^{2}\;.$$

## 4. Results

### 4.1. Baselines and Evaluation Metrics

We use our model to solve a regression problem, to predict the minimum temperature in all the nodes, and a classification problem, to predict two classes Frost and No Frost. In case of regression, we compare with the following forecasting methods:Non-deep learning methods: historical average (HA), ARIMA with Kalman filter (ARIMA), vector auto-regression (VAR), and support vector regression (SVR). The historical average accounts for weekly seasonality and predicts for a day by using the weighted average of the same day in the past few weeks;Deep learning methods that produce forecasting for each series separately (not graph-based) such as feed-forward neural network (FNN) and LSTM;Autoencoder forecasting method with attention mechanism (AC-att);Graph convolutional network applied to the given graph without spatio-temporal attention mechanism (GCN);Variants of this architecture using convolutions [7] and GRU [26].

In case of classification, we do not use the non-deep learning methods described above, but we use support vector machines (SVM) and tree-based classification algorithms such as naïve Bayes and XGBoost.

For regression, all methods are evaluated with three metrics: mean absolute error (MAE), root mean square error (RMSE), and mean absolute percentage error (MAPE). For classification, all methods are evaluated with precision, recall, and F1 metrics.

### 4.2. Hyperparameters

Several hyperparameters are tuned through grid search: initial learning rate in $\left\{0.1,0.01,0.001\right\}$, dropout rate in $\left\{0.1,0.2,0.3\right\}$, embedding size of the LSTM layer was set in $\left\{32,64,128,256\right\}$, the *k* value in kNN in $\left\{5,10,20,30\right\}$, and the weight of regularization in $\left\{0,1,2,5,10,20\right\}$. For other hyperparameters, the convolution kernel size in the feature extractor is 10 and the decay ratio of learning rate is 0.1. After tuning, the best initial learning rate for our dataset is 0.001. The optimizer is Adam.

All models are implemented in PyTorch and ran in the Google Colaboratory platform.

### 4.3. Results for Regression Problem

Table 1 and Figure 6 show the evaluation of the proposed GSTA-RCN model compared with that of the baselines. The tasks are to forecast the minimum temperature of the experimental field with 6 h, 12 h, 24 h and 48 h in advance.

The proposed model outperforms all the compared baselines for frost forecast in 6 h, 12 h, 24 h and 48 h tasks. A nongraph model such as an autoencoder with an attention mechanism outperforms GNN with spatio-temporal attention using convolution and GRU. To improve these results, it is necessary to collect more weather data and weather variables and to use more weather stations for modeling geographical and temporal interactions.

In general terms, we can separate the results from non-deep learning models and FNN from LSTM, autoencoder and graph-based neural network architectures. Non-deep learning models (HA, ARIMA) and the simple deep learning architecture FNN only have similar error scores with the other architectures on the 6 h task. For a greater time-window, their performance drastically decreases. Then, we can compare results from LSTM with the results from STA-C, the variation of the spatio-temporal GNN with attention mechanism, and a convolution-based forecasting method. For this dataset, the complexity added by STA-C does not have an impact on the performance of the model and a simple LSTM is preferable, especially for 6 h, 12 h, and 24 h predictions. As mentioned previously, the autoencoder architecture with attention mechanism performs better than STA-C and similar to STA-GRU for 6 h, 12 h, and 24 h tasks. Finally, our model outperforms all the previous baselines. In our case, the complexity of the modeling successfully increases the performance of the prediction for each time-window task.

Figure 7 shows model’s results in terms of RMSE for the experimental field node and the three geographically closest node neighbors. By focusing on the prediction for each node, instead of the average of all nodes, the results description remains the same. In addition, the Pearson correlation statistic is calculated to compare model prediction with time windows. The performance of all models decreases when the time-window increases, and its results are variable in the different nodes. Therefore, the behavior of models regarding time-window is different in each node, which is a result that could be worth to continue studying.

In addition, Figure 8 and Figure 9 show real and predicted data for a specific day. In Figure 9 we present the variation of our model prediction regarding 6 h, 12 h, 24 h, and 48 h tasks. Our model’s best performance is in the 6 h window task, and then the performance decreases gradually. Although the performance decreases for larger time windows, our model is capable of detecting the temperature trend. The range of the error interval increases if we want to make the prediction point by point, but if that range is detected, we can be sure that the prediction is reliable. Figure 8 shows our model prediction and the second-best prediction (STA-GRU) in the 6 h task. Both models detect the temperature trend, but in our model the error range is smaller.

### 4.4. Results for Classification Problem

Table 2 and Figure 10 show the evaluation of the proposed GSTA-RCN model compared with the baselines as a classification problem. In this problem, two classes are predicted: Frost (temperature below 0 °C) and No Frost (temperature above 0 °C).

In this case our model also outperforms all the baselines for the 6 h, 12 h, 24 h, and 48 h tasks. In addition, similar to regression, models performance decreases when time-window increases.

For this dataset, Naive Bayes and SVM models provide the worst predictions, especially for recall score, which implies a high value of false-negative predictions. Compared with regression, in classification, FNN and LSTM perform similar. The autoencoder with attention mechanism is slightly better that the previous ones but performs worse than all the graph-based models and even the XGBoost model. Figure 10 shows that for classification in this dataset the graph-based models are preferable. In this case, the spatio-temporal attention mechanism in the GNN, which captures the dynamics between the nodes, is a key factor to improve the predictions. In particular, in our model the strategy to use the spatio-temporal vectors to perform the prediction at the same time obtains the best results by minimizing classification errors, which could be valuable for real-world applications.

## 5. Discussion

### 5.1. GNN and Spatio-Temporal Attention Applied to Frost Forecast

The first research question pretends to evaluate the performance of the GNN against current state-of-the-art models for frost forecasting. In other words, does the use of a deep learning model with graph-based data improve the performance of a predictor?

The overall results indicate that our model outperforms previous frost forecasting models and most of the variations presented for graph neural networks. To statistically check whether the models’ performance difference is significant, we conducted a *t*-test between our model and each previously proposed frost forecasting model and other baselines presented. Table 3 and Table 4 shows the statistics results for regression and classification, respectively.

As can be seen in Table 3 and Table 4, there is statistical significance for the results of our model against all previously proposed frost forecasting models for 6 h, 12 h, 24 h, and 48 h in advance predictions. More generally, for all the baselines presented, there are some exceptions in the 48 h time-window prediction since there is no evidence of a difference between our model and STA-GRU. This result can be used for future research with the purpose of improving the performance in larger time-windows.

The second research question “Does the spatio-temporal attention mechanism improves the forecast?” is approached similarly to the previous one. In this case, we are interested to statistically check whether there is a difference between the performance of our model with GNN models that do not use spatio-temporal attention mechanisms. In particular, Table 5 shows the results of a *t*-test applied between our model, GCN_reg_ and GCN_clas_ separately for regression and classification problems. As a result, we can demonstrate that for our dataset, results are statistically different between our model and non-spatio-temporal attention graph-based models.

The last research question, regarding whether the combined use of different data sources improve the forecast in comparison with using a single data source, can be answered based on the first two questions. By answering the two first research questions, we are implicitly answering the last one since the use of different data sources imply on the use of weather data collected from different locations. Therefore, it is possible to construct the graph and collect the spatial relationships between the nodes, which is crucial for the model’s performance.

Finally, in terms of an evaluation for the IoT-based system in comparison with the weather stations used, as shown in Figure 7 the overall results indicate that there is no difference between sites. Concretely, Table 6 presents statistics by comparing the prediction results of using our model in the experimental field against each weather station. There is no statistical significance that indicates a performance difference. Therefore, data collected from our system are reliable.

### 5.2. Limitations

The main limitation of this model is concerned with the decrease in performance with larger time-windows. Despite that it is a problem for all the models studied in this paper, the performance difference of our model in short-term windows with all models cannot be established in 48 h time-windows. We suspect that this behavior is similar for even larger time-windows. The lack of statistical significance, in those specific cases, is probably a consequence of a constrained period of time in which we collected the data from the experimental site and weather stations. Consequently, model performance should be revisited when the amount of data collected is larger, one year and more. However, deeply studying the deep learning architecture used and investigating whether further improvements can be made to obtain better results for long-term predictions are of interest.

## 6. Conclusions

We presented our frost forecasting model that uses and optimizes a graph structure between multiple time series using a graph neural network (GNN) architecture with a recurrent graph convolution mechanism to process each series simultaneously. The model concludes with spatio-temporal attention to consider spatial relations and extract temporal dynamics.

Frost forecast is an important area of climate research because of its economic impact on several industries. In this study, a GNN with spatio-temporal architecture was proposed to predict minimum temperatures at an experimental site. The model considers spatial and temporal relations and processes multiple time series simultaneously. Performing predictions of 6, 12, 24, and 48 h this model outperforms statistical and nongraph deep learning models.

To further improve this model, we will continue our research by studying deep learning architectures to specifically adapt to different time-window forecasts. In addition, we aim to include domain knowledge from climate sciences that could help in the construction of the graph to transit from a statically defined graph to a dynamically defined one. Finally, by including domain knowledge or by applying new methods, we want to extract the influences of the nodes with each other for the purposes of explaining the dynamics of the graph and, as a consequence, to provide better practical insights to users of the system.

## Figures and Tables

**Figure 1 sensors-22-01486-f001:**
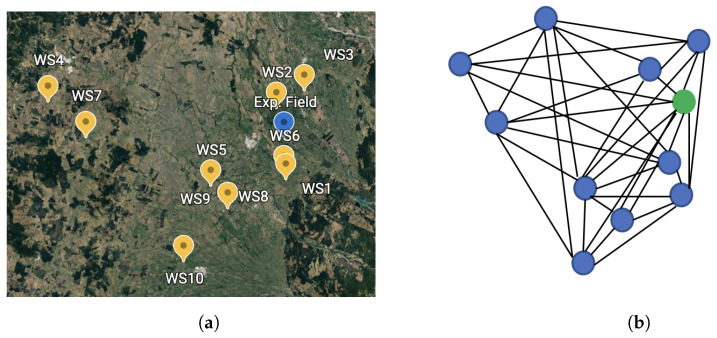
Graph modeling based on location of experimental site and weather stations. (**a**) Map of weather stations and experimental site; (**b**) Schematic version of a graph constructed at a single time step.

**Figure 2 sensors-22-01486-f002:**
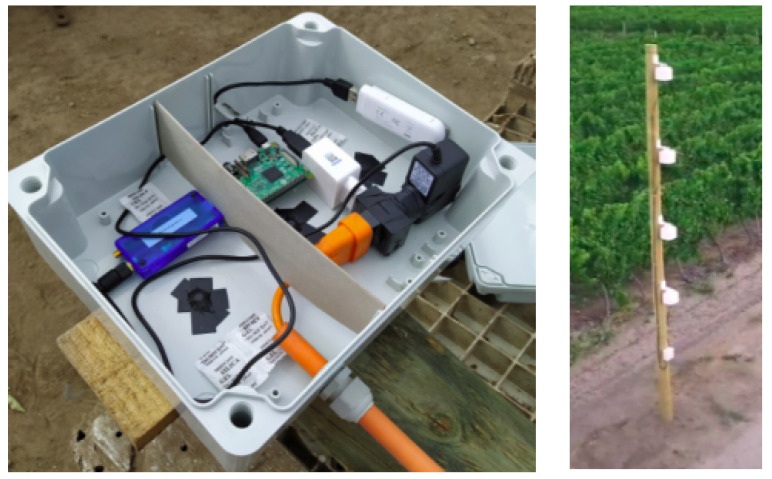
Internet of things (IoT) platform installed on experimental field composed by temperature and humidity sensors.

**Figure 3 sensors-22-01486-f003:**
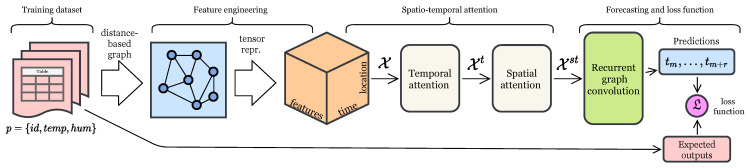
Overview of GRAST-Frost model.

**Figure 4 sensors-22-01486-f004:**
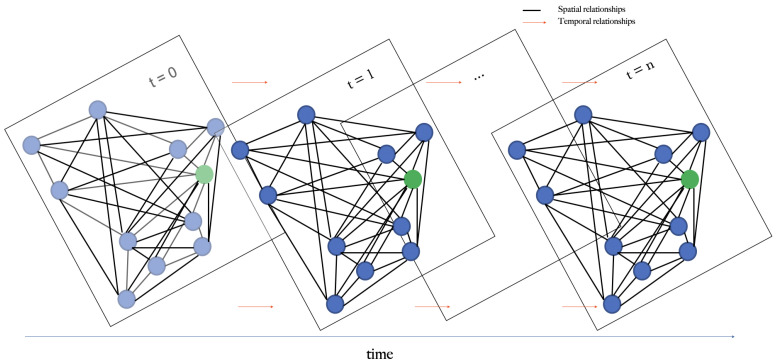
Schematic view of spatio-temporal graph network for modeling relationships between experimental field and weather stations through time.

**Figure 5 sensors-22-01486-f005:**
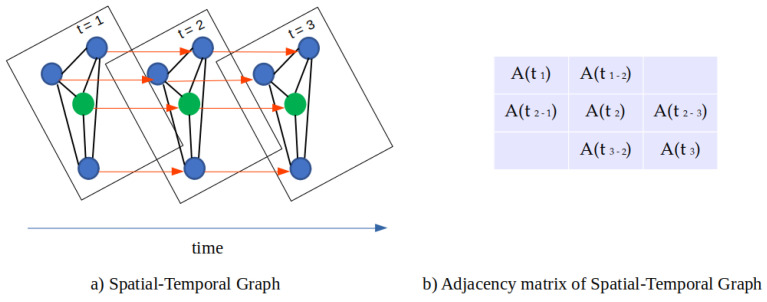
Spatio-temporal graph construction and adjacency matrix for including multiple time steps.

**Figure 6 sensors-22-01486-f006:**
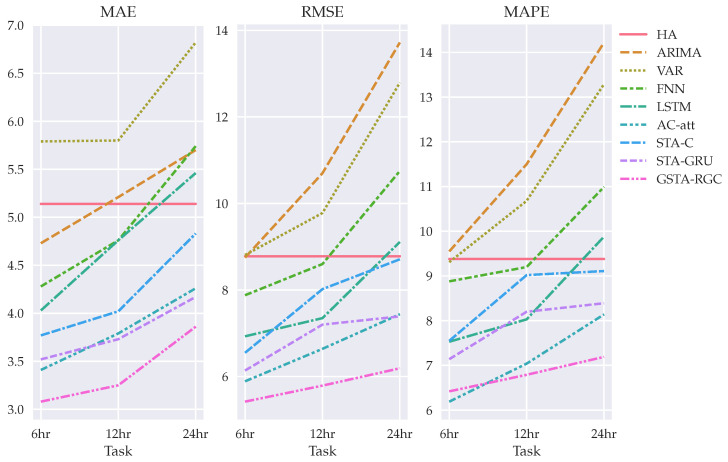
MAE, MAPE, and RSME metric results for temperature forecasting on 6, 12, and 24 h in advance.

**Figure 7 sensors-22-01486-f007:**
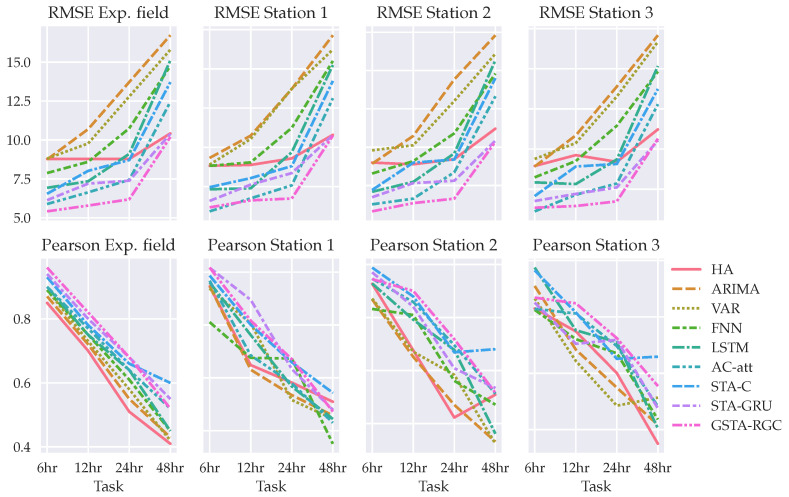
RMSE and Pearson statistic results regarding forecasting on the experimental field and three nearest weather stations.

**Figure 8 sensors-22-01486-f008:**
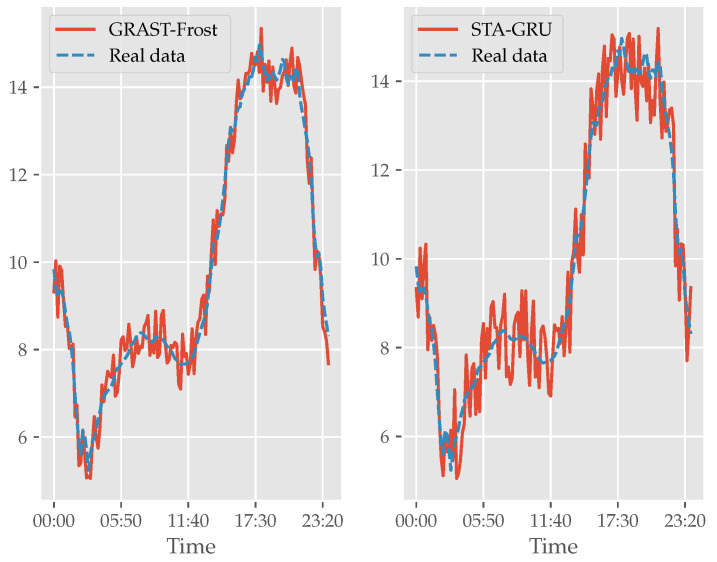
On left, there is a comparison of GRAST-Frost model with real data; on right, a comparison of STA-GRU model with real data.

**Figure 9 sensors-22-01486-f009:**
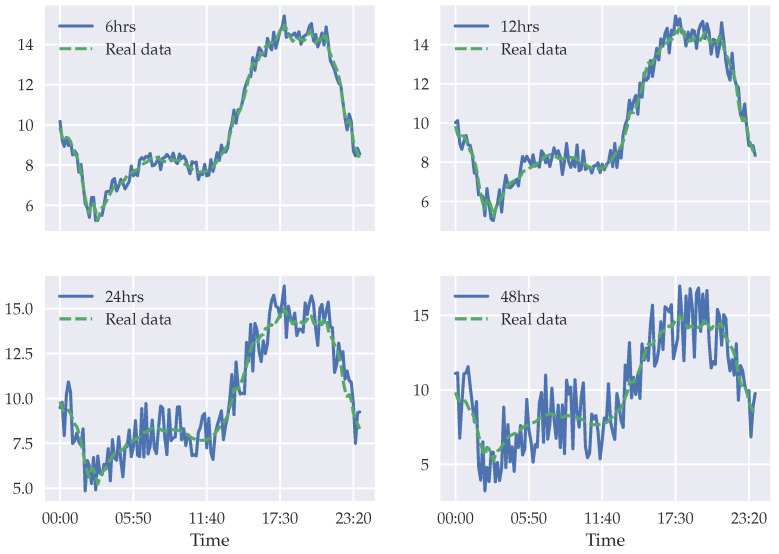
GRAST-Frost model results compared with that of real data for 6, 12, 24, and 48 h forecasts.

**Figure 10 sensors-22-01486-f010:**
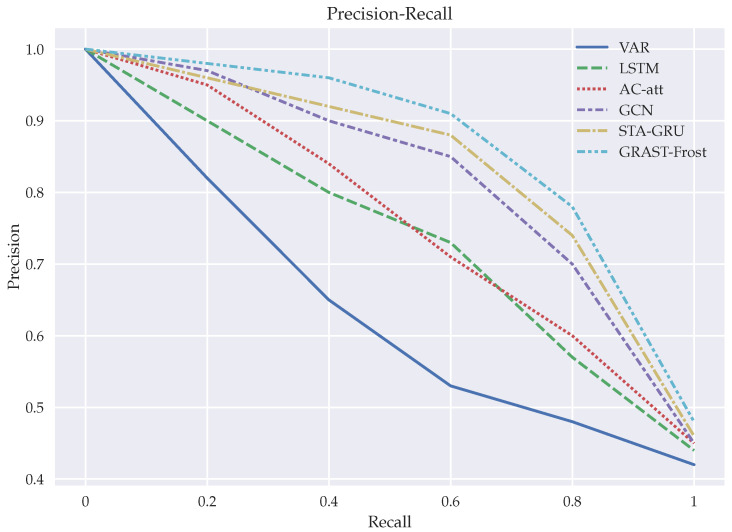
Receiver operating characteristic (ROC) curve comparing GRAST-Frost with other models implementations.

**Table 1 sensors-22-01486-t001:** Average mean absolute error (MAE), root mean square error (RMSE), and mean absolute percentage error (MAPE) metrics of time series forecasting models applied to frost forecast. Best values are highlighted in bold.

	MAE			RMSE			MAPE		
**Model**	**6 h**	**12 h**	**24 h**	**6 h**	**12 h**	**24 h**	**6 h**	**12 h**	**24 h**
VAR FNN	4.28	4.76	5.74	7.88	8.60	10.75	7.21	8.77	9.98
LSTM	4.03	4.76	5.46	6.93	7.35	9.11	6.95	6.89	8.61
AC-att	3.41	3.79	4.26	5.89	6.64	7.44	5.74	5.84	7.05
GCN	3.61	3.89	4.28	6.01	7.09	7.46	5.55	5.89	7.03
STA-C	3.77	4.02	4.83	6.55	8.02	8.71	6.11	7.87	8.30
STA-GRU	3.52	3.73	4.17	6.14	7.20	7.39	5.82	5.99	7.05
GRAST-Frost	**3.08**	**3.25**	**3.86**	**5.42**	**5.79**	**6.19**	**5.39**	**5.44**	**6.04**

**Table 2 sensors-22-01486-t002:** Average precision, recall and F1 metrics for classification of frost events according different models implementations. Best values are highlighted in bold.

	Precision			Recall			F1		
**Model**	**6 h**	**12 h**	**24 h**	**6 h**	**12 h**	**24 h**	**6 h**	**12 h**	**24 h**
SVM	0.569	0.535	0.513	0.565	0.529	0.507	0.555	0.531	0.502
XGBoost	0.754	0.740	0.713	0.749	0.730	0.701	0.748	0.716	0.711
FNN	0.644	0.643	0.638	0.663	0.658	0.639	0.690	0.669	0.658
LSTM	0.675	0.666	0.652	0.695	0.650	0.632	0.682	0.647	0.641
AC-att	0.713	0.709	0.690	0.705	0.700	0.696	0.699	0.691	0.671
GCN	0.792	0.787	0.780	0.784	0.778	0.769	0.806	0.802	0.787
STA-C	0.814	0.797	0.794	0.837	0.810	0.809	0.830	0.826	0.784
STA-GRU	0.845	0.844	0.824	0.850	0.821	0.805	0.863	0.845	0.842
GRAST-Frost	**0.891**	**0.876**	**0.848**	**0.898**	**0.882**	**0.845**	**0.908**	**0.869**	**0.863**

**Table 3 sensors-22-01486-t003:** *t*-test for comparing our regression model with baselines for 6 h, 12 h, 24 h and 48 h forecasts.

	6 h		12 h		24 h		48 h	
**Model**	*t*	* **p** * **-Value**	*t*	* **p** * **-Value**	*t*	* **p** * **-Value**	*t*	* **p** * **-Value**
HA	9.662	0.011	7.716	0.022	7.398	0.028	7.276	0.031
ARIMA	9.673	0.012	9.443	0.026	8.534	0.0036	8.139	0.033
VAR	10.330	0.017	8.247	0.041	7.831	0.045	7.172	0.044
FNN	5.946	0.020	5.722	0.025	5.595	0.030	4.239	0.041
LSTM	6.540	0.029	6.260	0.024	5.144	0.039	4.136	0.045
AC-att	4.293	0.017	3.674	0.020	3.562	0.025	3.179	0.035
STA-C	3.314	0.011	2.980	0.014	2.593	0.019	2.284	0.041
STA-GRU	1.387	0.030	1.271	0.038	1.009	0.041	0.766	0.081

**Table 4 sensors-22-01486-t004:** *t*-test for comparing our classification model with baselines for 6 h, 12 h, 24 h, and 48 h forecasts.

	6 h		12 h		24 h		48 h	
**Model**	*t*	* **p** * **-Value**	*t*	* **p** * **-Value**	*t*	* **p** * **-Value**	*t*	* **p** * **-Value**
SVM	9.610	0.015	9.042	0.036	7.946	0.038	7.939	0.049
NB	9.527	0.023	8.129	0.032	7.139	0.045	7.051	0.046
XGBoost	3.786	0.013	3.712	0.022	3.467	0.041	3.378	0.043
FNN	6.765	0.016	6.002	0.026	4.758	0.030	4.206	0.036
LSTM	6.095	0.018	5.245	0.018	4.586	0.026	4.488	0.046
AC-att	4.886	0.025	4.398	0.029	3.665	0.034	3.479	0.040
STA-C	3.018	0.023	2.579	0.029	2.152	0.047	2.084	0.048
STA-GRU	1.476	0.039	1.094	0.036	0.986	0.047	0.713	0.071

**Table 5 sensors-22-01486-t005:** *t*-test for comparing our model with graph-based models without spatio-temporal attention for 6 h, 12 h, 24 h and 48 h forecasts.

	6 h		12 h		24 h		48 h	
**Model**	*t*	* **p** * **-Value**	*t*	* **p** * **-Value**	*t*	* **p** * **-Value**	*t*	* **p** * **-Value**
GCN_reg_	3.631	0.013	3.412	0.018	2.456	0.033	2.416	0.035
GCN_clas_	4.009	0.012	3.861	0.031	2.859	0.032	2.787	0.041

**Table 6 sensors-22-01486-t006:** *t*-test for comparing our model forecast results from experimental site with weather stations.

	6 h		12 h		24 h		48 h	
**Model**	*t*	* **p** * **-Value**	*t*	* **p** * **-Value**	*t*	* **p** * **-Value**	*t*	* **p** * **-Value**
WS1	0.963	0.075	0.810	0.070	0.764	0.098	0.665	0.093
WS2	0.843	0.071	0.683	0.061	0.629	0.069	0.540	0.084
WS3	0.802	0.075	0.676	0.075	0.568	0.081	0.501	0.083
WS4	0.821	0.064	0.731	0.064	0.691	0.078	0.593	0.088
WS5	0.915	0.09	0.025	0.102	3.921	0.136	4.326	0.119
WS6	0.817	0.111	0.660	0.081	0.607	0.105	0.535	0.116
WS7	0.930	0.087	0.703	0.099	0.669	0.092	0.653	0.099
WS8	0.915	0.078	0.025	0.083	3.921	0.118	4.326	0.139
WS9	0.924	0.060	0.918	0.086	0.740	0.111	0.723	0.118
WS10	0.903	0.094	0.794	0.097	0.741	0.126	0.602	0.137

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
