# Peer review of "A Graph Neural Network with Spatio-Temporal Attention for Multi-Sources Time Series Data: An Application to Frost Forecast†"

_sensors, 2022, doi:10.3390/s22041486_

Round 1

Reviewer 1 Report

The paper is quite OK, but the proposal must be explained so much better.

You must explain better Figures 3 and 5 i.e., the temporal relation must be explained so much better.

The section 3.3.3 must be explained better, and also maybe put a graphic representing these layers: the temporal attention and the spatial attention layer.

Also give a better explanation of the Recurrent Graph Convolutional network, how many convolutional layers, what kind of recurrent network is used, i.e., explain the architecture of the RGC network used as a final step or your model.  You use an autoencoder, a GRNN a LTSM, a combination of them, what?. Because, it is not possible that we have to read so many papers in order to understand this paper.

Have you optimize the parameters of all the methods considered in the comparison or are results collected from other papers in the literature applied to the same problem?. Because in the firs case you must put in the paper, the architecture used in each method with their parameters, and I suppose that you have optimized them in order to obtain the bests results. If the response is the second case you must put the references of each paper where there are the results.

For the optimization of your method, you have used a grid search, this is OK, but have you used cross-validation procedure to do that, or you uses always the same data to train and to test the results for the hyperparameters? You must specify this better.

Author Response

Thank you very much for your comments. We have made our best to improve the paper following your advice.

We will address now address your comments line by line:

Reviewer comment (RC): You must explain better Figures 3 and 5 i.e., the temporal relation must be explained so much better. The section 3.3.3 must be explained better, and also maybe put a graphic representing these layers: the temporal attention and the spatial attention layer.

Authors reply (AR): We have improved the explanations as you requested. Section 3 has been improved substantially to facilitate the understanding the model.

RC: Also give a better explanation of the Recurrent Graph Convolutional network, how many convolutional layers, what kind of recurrent network is used, i.e., explain the architecture of the RGC network used as a final step or your model.  You use an autoencoder, a GRNN a LTSM, a combination of them, what?. Because, it is not possible that we have to read so many papers in order to understand this paper.

AR: We have improved the explanation the model including all the supporting formulae in order to address your comment.

RC: Have you optimize the parameters of all the methods considered in the comparison or are results collected from other papers in the literature applied to the same problem?. Because in the firs case you must put in the paper, the architecture used in each method with their parameters, and I suppose that you have optimized them in order to obtain the bests results. If the response is the second case you must put the references of each paper where there are the results.

AC: We use the parameters of the other models as published by their respective authors.

RC: For the optimization of your method, you have used a grid search, this is OK, but have you used cross-validation procedure to do that, or you uses always the same data to train and to test the results for the hyperparameters? You must specify this better.

AC: We have used cross-validation. This has been more evident in the text.

Reviewer 2 Report

Original Submission

Recommendation

Major (very) revision

Comments to Author:

Title:  

A Graph Neural Network with Spatio-temporal Attention for Multi-sources Time Series Data: An application to Frost Forecast

Overview and general recommendation.

This paper is demonstrating usage of NN methodology and some important applications. This technique is very important and somehow accurate and has lots of important applications. First of all, as a person with more than 20 years familiarity with NN/AI/ML data, I like this paper very much; but as a scientist, I have to say the truth about the material and to be honest.

I think this paper came from a PHD thesis; am I right? If it is so, you cannot just bring it right to MDPI=> you have to work a lot on the paper and show us the skim of the work. In the current form I am not OK with this paper; unless you do your job and skim it.

The Abstract is OK: summarizing the idea and concepts inside the paper; English is a bit problem!! I think it is better to give it to a native person to review (it is a big must). I feel like the abstract is raw: pls think more about what you have done in the paper and do the abstract again.

I do not think that Introduction is fair. I think in some positions, some important corrections must be done (I think it’s better to write it again; but it is not a must!); leak of lots of Refs... please fix them! Pls improve the quality of Figs> they are very very bad; unacceptable (they look like a screen shot!!). I think related work (section2) must be merged with section 1; in that case section 1 would be huge-> so make it works.

Results, discuses, are needed to be reconsidering seriously.

I like this paper very much: good experiments have been done; however, I think this work must be improved and lots of things to do; agreed? I think the paper is very raw, and must be improved tremendously. At this stage, I think this paper is not suite for MDPI, so, pls do these primary corrections, then I will go over the paper again.

Detailed comments:

Line 123: why did not introduce IOT at the first write?

Lines 212-213. Redundancy => remove this paragraph pls.

Line 228. RxR=> are you kidding? Put a multiplication sign!!!!!

Fig.1. put a) and b) and explain EACH!

Fig.2. is very bad: replace it with better map; I cannot even read the names!

Lines 257-266. Do we need all these coordinates to see?

Fig.4. is very unprofessional!!

- Quality of the Figs are disaster!!!

- you just put one table in one page!!! (page11).

Fig.8. explain Figs more!!

Author Response

Thank you for your review and comments. We have addressed them and produced an improved version of the paper.

We will go over your comments now and detail how we have addressed them.

Reviewer 2 comment (RC2): This paper is demonstrating usage of NN methodology and some important applications. This technique is very important and somehow accurate and has lots of important applications. First of all, as a person with more than 20 years familiarity with NN/AI/ML data, I like this paper very much; but as a scientist, I have to say the truth about the material and to be honest.

Authors reply (AR): Thank you for acknowledging the impact of our work. 

RC2: I think this paper came from a PHD thesis; am I right? If it is so, you cannot just bring it right to MDPI=> you have to work a lot on the paper and show us the skim of the work. In the current form I am not OK with this paper; unless you do your job and skim it.

AR: We don't know how you got to this conclusion but it is incorrect. Anyhow, we understand your point on improving and polishing the overall presentation and discussion of results. We belief that doing that, we have address your concern.

RC2: The Abstract is OK: summarizing the idea and concepts inside the paper; English is a bit problem!! I think it is better to give it to a native person to review (it is a big must). I feel like the abstract is raw: pls think more about what you have done in the paper and do the abstract again.

AR: We have improved the abstract correspondingly.

RC2: I do not think that Introduction is fair. I think in some positions, some important corrections must be done (I think it’s better to write it again; but it is not a must!); leak of lots of Refs... please fix them! Pls improve the quality of Figs> they are very very bad; unacceptable (they look like a screen shot!!). I think related work (section2) must be merged with section 1; in that case section 1 would be huge-> so make it works.

AR: We have have worked a lot on the quality of the text, plots and figures. We have reorganized the text. However we disagree with your suggestion of merging Sections 1 and 2 as it would confuse the readers. We did improve the text to make the text easier to follow. 

RC2: Results, discuses, are needed to be reconsidering seriously.

AC: We have improved the representation of results accordingly.

RC2: I like this paper very much: good experiments have been done; however, I think this work must be improved and lots of things to do; agreed? I think the paper is very raw, and must be improved tremendously. At this stage, I think this paper is not suite for MDPI, so, pls do these primary corrections, then I will go over the paper again.

AR: In our humble opinion we have substantially improved the paper with better text, figures and plots. We have also sent the paper to a professional language service for analysis.

Detailed comments:

RC2: Line 123: why did not introduce IOT at the first write?

AR: We don't fully understand the question. If you are asking why we didn't introduce the IoT setup earlier it is because the paper is focused on the ML model and not on IoT.

RC2: Lines 212-213. Redundancy => remove this paragraph pls.

AR: Thanks, fixed.

RC2: Line 228. RxR=> are you kidding? Put a multiplication sign!!!!!

AR: Fixed.

RC2: Fig.1. put a) and b) and explain EACH!

AR: Fixed

RC2: Fig.2. is very bad: replace it with better map; I cannot even read the names!

AR: New and improved figure.

RC2: Lines 257-266. Do we need all these coordinates to see?

AR: We moved them to an appendix.

RC2: Fig.4. is very unprofessional!!

AR: New figure with high quality.

RC2: - Quality of the Figs are disaster!!!

AR: We have now high-quality figures.

RC2: - you just put one table in one page!!! (page11).

AC: Fixed

RC2: Fig.8. explain Figs more!!

AC: Better explanation is provided in text.

Round 2

Reviewer 2 Report

The authors have been responded almost all of the comments, and seems to me they are in good understating of the material; the paper is improved remarkably, and in this form, I am positive to accept that,

Good Luck!